# Systematic Child Talks in Early Childhood Education—A Method for Sustainability

**DOI:** 10.3390/children10040661

**Published:** 2023-03-31

**Authors:** Ingrid Engdahl, Ingrid Pramling Samuelsson, Eva Ärlemalm-Hagsér

**Affiliations:** 1School of Education, Culture, and Communication, Mälardalen University, 72123 Västerås, Sweden; 2Department of Education, Communication and Learning, University of Gothenburg, 40530 Gothenburg, Sweden; ingrid.pramling@ped.gu.se

**Keywords:** early childhood education, sustainability, systematic child talks, child interviews, teacher competence, preschool, early childhood education for sustainability

## Abstract

One of a preschool teacher’s most important competencies is to be able to talk with children and to invite them to share their ideas, knowledge, and experiences. This skill is of utmost importance within Early Childhood Education for sustainability. The aim of this article is to show various ways in which preschool teachers carry out systematic talks with children. Data come from a large Swedish development and research project, Sustainable Preschool, involving around 200 teachers in Early Childhood Education. During the spring of 2022, the preschools carried out theme-oriented projects linked to sustainable development. The participating preschool teachers were then asked to carry out systematic child talks with children about learning for sustainability and their understanding of the sustainability-related content. Using content analysis, three different approaches were identified as to how teachers communicate with children systematically about various content related to sustainability: (1) joint creation of meaning, (2) question and answer, focusing on remembering facts, and (3) following the children. There is a large variation in the teachers’ communicative competences. A key factor seems to be to create a shared inter-subjective atmosphere, while at the same time being open for alterity, that is, introducing new or slightly changed perspectives for the dialogue to deepen and continue.

## 1. Introduction

One of a preschool teacher’s most important competences is to be able to talk with children and to invite them to share their ideas, knowledge, and experiences. This article aims to provide further knowledge on Early Childhood Education for Sustainability (ECEfS) [1], especially the need for studies on how communication in early childhood education (ECE) may promote handling issues of sustainability in ECE. The aim of this article is to show various ways in which preschool teachers carry out systematic talks with children in preschool, with data coming from a development and research project, Sustainable Preschool run by the IFOUS Institute (Innovation and research development in school and preschool) [2].

In this study, the focus is on systematic child talks with children about sustainability issues in ECE. Research shows that changing the world to a more sustainable place leans on communication and democracy—that all people must be able to express themselves as citizens, listen to others, and negotiate meaning jointly [3,4]. This is why communicative skills are central for a re-orientation towards handling sustainability issues.

Moss [5] claims that there is a possibility for ECE to be a place of democratic political practice. He points out: “When I say that there is a possibility that institutions for children and young people can be, first and foremost, places of democratic political practice. I say ‘possibility’, to emphasize that this understanding is a choice that we, as citizens, can make” (p. 1, [5]). Further on, Moss (p. 3, [5]) lists why democratic practice is so important: (i) democratic participation is an important right to citizenship; (ii) democracy is the best defense against totalitarianism, whether in government or other institutions; (iii) democracy creates the possibility for diversity to flourish, which would be the best environment for the production of new thinking and practice. In other words, the idea that democracy is a form of living together must be honored and spread. 

Democracy, then, is a way of being and of thinking of oneself, of others, and of the world as deeply related. Democracy may give maximum opportunities for sharing and exchanging perspectives and opinions. In our times, democracy and sustainability represent both fundamental educational values and govern the form of educational activity. The above reasoning is the rationale when we argue in this article that communication, for example, children–teacher communication, child talks, and promoting children’s talking, is a method for education for sustainability (see also [6]). Jickling and Sterling state that the times we live in call for a brave, transformative, and conscious approach in education to manage the necessary re-orientation of all education towards sustainability [7]. 

The purpose of this study was to explore the ways in which preschool teachers tackle the task of conducting systematic child talks in their practices, focusing on education for sustainable development. These research questions guided us through the study:What links can be found between the preschool teachers’ verbal approaches in the talks and different views on children and knowledge?How are the different approaches to knowledge related to education for sustainability? 

## 2. Materials and Methods

This chapter starts with a broad overview of research in the field in Section 2.1, Section 2.2 and Section 2.3, followed by a presentation of ECE in Sweden in Section 2.4, and the theoretical framework of the study in Section 2.5. The final Section 2.6 presents methods and context, including ethical considerations.

### 2.1. Understanding Early Childhood Education Politically—A Broad Perspective of Variation

On a state level, the mission, the access, and the conditions for early childhood education and care (hereafter, ECEC) are the responsibility of the government. The UN Sustainable Development Goal, target 4:2, states “By 2030, ensure that all girls and boys have access to quality early childhood development, care and pre-primary education so that they are ready for primary education” [8]. The international community is forced to step up to reach the SDG, and heads of states are negotiating how to reach the goals, for instance, in the recent UN Tashkent Declaration on ECCE [9]. Additionally, target 4.7 spells out the content for teaching children about sustainability, which is the focus of this study. The SDGs challenge the provision of possibilities for ECEC to develop in a democratic approach and for everyday life. However, there are many different approaches to ECEC; thus, UNESCO-led global discussions on strategic partnerships between all parts of society are necessary in the best interests of young children [10]. 

When it comes to views on knowledge and on children, we can see that the years before formal schooling are labeled differently in different countries. Not only are the activities for children called different things, but also, large global organizations use different concepts and acronyms. UN and UNICEF call activities before primary school Early Child Development (ECD), which signals development as the main task. UNESCO, another large player in the area, uses Early Childhood Care and Education (ECCE), which signals that care is more important than education [11]. 

Finally, if we look at the ECE research community and the European Union, the most usual way to talk about this age group is Early Childhood Education (ECE), sometimes with a clarifying—and Care—attached (ECEC), which signal a more active view on learning and the right to education, as stated in the UN Convention on the rights of the Child [12]. These notions show that children’s own experiences, the holistic view on education and care, and children’s right to participate in their own learning process are valued differently, whereby some agencies stress child development and others highlight education, educare, or education and care. We can also see in a comparative policy study of majority and minority world countries how education and care are related to sustainable development in different countries in the majority and minority world [13]. Some countries divide ECEC for young children by age, where ECEC for children under a certain age are focusing on care. 

### 2.2. Young Children as Social, Active, and with Competences

The everyday life of young children and the view on children and children’s competences has changed a lot during the last 50 years. In the book The first 1000 days of early childhood: Becoming, Gradovski and colleagues [14] point out the biological base that makes early childhood a unique period in one’s life span. The young child also lives in a world that is culturally mediated and dialogically arranged according to local circumstances. Children will be influenced by the preference of others around them and the consequences of certain values or beliefs, both locally and nationally constituted [14]. The authors use an example about how the view of babies has changed towards a convincing idea that young children need each other as peers, not only caregivers. Today, children in many countries spend time from their early years in early childhood education and care settings, which in Sweden and in this article are called preschools. In 2021, 50 percent of one-year-olds, 93 percent of two-year-olds, 95 percent of three-year-olds, and 96 percent of four- and five-year-olds participated in ECEC [15]. Thus, as preschool teachers are close to children daily, the ways in which they act, think, and communicate with children will influence play, development, and learning.

Established in the 1980s, research on the development of infants and toddlers changed from mainly laboratory experiments towards studies in natural environments, in homes and preschools, made possible by the use of cameras and video recordings. This methodological change resulted in a new way of looking at children’s ways of acting, reacting, behaving, and learning [16]. Research with infants and toddlers in the Nordic countries was early in studying natural environments in preschool settings [1]. Studies of infant development show that a child can be socially active from the first minutes of its life, and that interaction take place in meaningful contexts [17,18]. 

Stern [18] used video recordings when studying the youngest children and the notion of intersubjectivity. He developed a theoretical approach—the interpersonal world of the infant. Children have an emergent sense of self from birth, and they develop more senses (in Stern’s terminology, a core sense, a subjective sense, and a verbal sense) during interpersonal lifelong processes of communication. Competences such as emotional attunement, interplay, co-existence, and mutuality characterize emerging intersubjective and communicative skills already during the first years, the first 1000 days. Stern also shows how important “present moments’” of intersubjectivity [19] are for children’s communication, learning, and development. 

Following the above examples, more theories show that communication and interaction play a large role in children’s play and learning. In theories such as Vygotskij’s theory of mediation [20], in Developmental Pedagogy [21], or in Play-responsive teaching [22], communication is central.

### 2.3. Dialogues with Children: The Importance of a Competent Teacher

Trevarthen and Aitken showed the importance of communication between adults and children at very young ages [23]. This group of researchers showed how the very young child mirrored the face of the adult, and how communication changed depending on the mother’s mode. These results were only possible to recognize with new technology. Quality in ECE is closely related to the competence of the teachers and other staff [24]. Formal competence is not a guarantee for quality teaching. 

Communication can be understood as communicative actions, listening, speaking, and interpreting. Listening to children as an important aspect of their development and learning was first highlighted by Rousseau in 1762. He stressed that the adult’s role as a listener is to acknowledge the child as a communicative being. The adult must recognize the importance of a receptive and compassionate listening and verbal response towards children [25]. Communication can be seen as monologic, a transmission of messages between adults and children, or as dialogic, in which communication is understood as a co-creation that contains both consensus, reciprocity, and alterity, thus inviting different perspectives and understandings in dialogues [26]. However, teaching is always a negotiating dialogue between the teacher and the child [27].

Johansson [28] observed communication and acting in praxis, taking the new knowledge about children’s intersubjective competences as a starting point. Her phenomenological analysis resulted in three main themes. The results indicate that an interactive atmosphere characterized by proximity to the child’s lifeworld is often accompanied by a view of the child as a fellow human being and a confidence in the child’s capacity to learn. In an unstable atmosphere, the view of the child is characterized by a perspective from above and a view of learning based on maturity and the child’s lack of competence. Finally, in a controlling atmosphere, a view of the child as irrational together with a view of learning based on conditioning was dominant [28]. These findings highlight that quality ECE varies with the teachers’ view on children, the way they look at knowledge and learning, and how these different categories become visible in their approaches to teaching and learning in ECE.

Eriksen Ødegaard [29] describes how a collaborative and exploratory working method presupposes that one is knowledgeable in having a dialogue with children and in steering towards issues of sustainability. Therefore, the ability to both be open and to steer the dialogue towards issues of importance for sustainability becomes crucial. Ärlemalm-Hagsér and Engdahl [30] show how communication and creativity contribute to children’s development of knowledge about the living conditions of eggs and chickens, something that resulted in changed purchasing routines for the preschools in their municipality. In a study of children’s understanding of participating in actions linked to Earth hour, Ärlemalm-Hagsér [31] highlights that the content of knowledge changes in the dialogue through negotiations between the children and between children and adults. Borg [32] used conversations with children in her research on how children think about finances and sustainability and describes that children exhibit varied and empathic perceptions. Hedefalk and colleagues [33] believe that if children do not get the opportunity to make their, or others’, voices heard, they do not gain the experience of problematizing conversations with critical elements in their education for sustainability.

In a large OMEP-study in 2010 on children’s voices about sustainability, two conclusions were drawn from the nearly 10,000 informal interviews with children: children are interested in a sustainable future, and they know more about these things than the interviewing teachers thought prior to the interviews [34]. 

### 2.4. Early Childhood Education in Sweden: Communication and Sustainability in Focus

In Sweden, the mission of ECEC is spelled out in the national curriculum [35]. The preschool is part of the school system and rests on the basis of democracy. The Education Act [36] stipulates that the purpose of education in the preschool is to ensure that children acquire and develop knowledge and values. It should promote all children’s development and learning, as well as a lifelong desire to learn. Education should also convey and establish respect for human rights and the fundamental democratic values on which Swedish society is based. The Swedish National Curriculum for the Preschool [35] states that language, learning, and the development of identity are closely linked. Some of the curricular goals to strive for are related to the state of communication that the preschool should provide each child with the conditions to develop:

“An ability to use and understand concepts, see correlations, and discover new ways of understanding the world around them;

An ability to create and an ability to express and communicate occurrences, thoughts, and experiences in different forms of expression such as image, form, drama, movement, singing, music, and dance”.(p. 15, [35])

These goals highlight communication both as important for challenging children’s learning via communication, as well as using communication for evaluating children’s understandings of something. Being able to hold conversations with children about different content is highlighted in the preschool curriculum [35]. Children’s development of language and agency in the area of sustainability is a key question in the Swedish national curriculum. The preschool should provide children with the conditions to develop:

“A growing responsibility for and interest in sustainable development and active participation in society”.(p. 13, [35])

“An understanding of relationships in nature and different cycles in nature, and how people, nature, and society affect each other;

An understanding of how different choices people make in everyday life can contribute to sustainable development;

An understanding of natural sciences, knowledge of plants and animals, and simple chemical processes and physical phenomena”.(p. 15, [35])

In other words, communication influences children’s learning and meaning making of the world around them and gives opportunities for the teacher to learn about the child’s learning and where they are in the process of meaning making linked to sustainability. Thus, democratic negotiation and communication skills are necessary, both to influence the child’s world and for the teacher to make sense of the child’s world. 

### 2.5. Theoretical Perspective

It is not possible to communicate with children without communicating about something. Developmental Pedagogy [37], referred to in ref. [21], is an approach based on children’s perspectives and meaning making as a learning process. The *what* aspect, called the “learning object”, is in focus for learning and communication. The content in focus could be aspects related to sustainable development, and the learning object is what is focused on in each specific situation and communication. In education and learning, there is also a *how* aspect, called the “learning act”, describing how the learning activities and situations are framed and promoted. In this article, we look at the learning act in which the systematic child talks and how they were carried out.

Developmental pedagogy looks at learning as a matter of being able to understand or express oneself in a qualitatively new way, or in other words, that children’s knowledge expands (develops) when they learn something [37]. To be able to follow the learning, it becomes necessary to try to understand children’s subjective world, both before and after the preschool teacher has worked with a specific piece of content for children to create an understanding around. It is about trying to understand the meaning the children create about the specific content, in communication with other children and adults in the preschool context [24,37], not whether the adult’s knowledge has been transferred to the child’s knowledge.

Communication with young children in ECEC requires professional competence, in which systematic child talks are a key for understanding and a shared sustainable thinking [4,24]. During the talks, children can express opinions, viewpoints, and ideas and develop ideas for agency in their everyday lives.

Research shows that interviewing children is different from interviewing adults [38,39,40]. Adults usually talk, and you only have to listen, but it really does not work that way when starting a dialogue with children. A more active approach is required. Above all, absolute attention is necessary, an ability to take the child’s perspective, and a sensitivity to when the child thinks enough is enough. Children like to express it as “are we done now?” or “can I go now?” It takes some practice to get children to share their world and to challenge children to further develop their thoughts [16,22,37].

In this article, the developmental pedagogy theory and theories about communication are framing the what and the how of doing systematic child talks. In research, the term child interview is often used for this type of systematic child talks. Such research interviews are characterized by asking open-ended questions to gain access to children’s subjective world. With systematic child talks, we do not mean conversations in general, but specific talks in which the intention is to approach children’s perspectives. It is about what children express as a result of their meaning makings about aspects of sustainability that the preschool has worked with. The content varies depending on which themes and topics the ECE has focused on [41].

### 2.6. Methods and Context

The study is part of a larger research and development program, Sustainable preschool, led by the Ifous Institute [2]. The program includes four municipalities, four districts in a larger city, and a national provider of preschool education. The participants in this study were 200 preschool teachers. During the spring of 2022, the preschools carried out theme-oriented projects linked to sustainable development. At the end of the semester, we asked the participating preschool teachers to talk with the children about something they completed work in the theme/project related to sustainability. The purpose of the preschool teachers’ task was to communicate with children about learning for sustainability and their understanding of the content addressed in the theme work. It was about trying to find out what the children perceived, noticed, and their meaning making of the content of the theme/project—in other words, children’s expressions of their subjective world. The participating preschool teachers were asked to transcribe and submit three child talks on a response form. These were then uploaded to a web platform with a secured login where only the researchers could access the material.

The task to do systematic child talks with children was introduced at a conference within the program for all participants. A presentation was given by one of the researchers about conducting systematic child talks and on ethical aspects connected to this. Systematic child talk is a method for approaching children’s perspectives by talking openly about children’s ideas, skills, or ways of thinking about a chosen subject [39]. Gaining access to the children’s world is always a matter of negotiation and dialogue, which in turn is based on creating a social contract. The preschool teacher needs to know what content he/she will focus on during the talk—and to be open to the utterances and answers. Furthermore, spontaneous talks were discussed in relation to staged talks, as were the difference between open and closed questions, and questions about technical equipment for child talks.

The Sustainable Preschool program applies high requirements for information about the purpose of the various sub-studies in the program [2]. All participants have given written consent to participate in the research within the program. Participation in the research and development program forms part of the preschool teachers’ employment, which provides good opportunities for transparency and influence of both research and development parts within the program. These prerequisites follow the ethical recommendations from the University of Uppsala [42] and the Swedish Research Council regarding confidentiality and dissemination of the results. Prior to the start of the study, an application for authorization was submitted to National Ethical Authority (dnr 2021-06472-01), and the study was started upon the response that no specific ethical trial was required.

#### Data and Analyses

We received a total of 399 transcribed response forms. The length of the interviews varied between 1–19 pages. Some preschool teachers had interviewed a smaller group of children, instead of individual children. In this article, we analyze the child talks between a teacher and one child, not the talks with groups of children. The focus of the analysis is on the dialogues between children and adults and on the children’s responses. 

The 399 talks were analyzed using interpretative content analysis [43]. We looked at the type of questions posed, whether they were open or closed, and what characterized the talks in terms of different teacher approaches. The analyses were first carried out individually by two of the researchers and were then compared in a second step. In this phase, three different approaches were identified. All three researchers participated in the discussions about the findings and the conclusions. The transcripts in this article were translated verbatim by one of the researchers. In the excerpts, T stands for Teacher, C stands for Child, and NN stands for the child’s name.

## 3. Results: Various Approaches in Systematic Child Talks between Preschool Teachers and Children

In the transcribed child talks, we could discern three different approaches of how teachers communicate with children systematically about various content related to sustainability. These have been labeled: (1) joint creation of meaning, (2) question and answer, focusing on remembering facts, and (3) following the children. These approaches are presented below and illustrated with six excerpts from the transcribed child talks. 

The starting point for the communication varies a lot. Teachers used different kinds of prompts, or just an invitational question to start up the communication. The prompts could be a photo, a collection of photos, a short video on a tablet, a book, or other forms of documentations.

### 3.1. Joint Creation of Meaning

In this approach, the child talks are characterized as a common activity with involved children. The climate is permissive and open, and the communication shows a shared focus. The preschool teachers include a learning orientation in their approach during the talk. In these child talks, we can see how the teacher and the child have a mutual dialogue, in which there is space and possibilities for the child to share his or her ideas, thinking, and the way in which they experience the content they are dealing with in the communication. The content is something they have worked with in preschool related to sustainability.

In the first excerpt, *Excerpt 1*. *Sharing ideas about animal protection*, we meet a six-year-old child during the first part of the interview.

Excerpt 1

T: if you think about what you feel you have learned about the animals during our project what do you say?C: that you shouldn’t litter in the forestT: what do you think is happening then?C: then the animals can eat the garbage and get the garbage in their throatsT: what could we do about it?C: when we go out into the forest the next time, maybe we should take a bag with usT: we could do that, but what if more people still come and litter? Can we do anything more to try to get everyone else to understand?C: make a sign!T: make a sign—we could absolutely do that. What would you like to write on it?C: greetings—no, “please don’t litter nature because animals can eat it and get litter in their throats and maybe die, please don’t—greetings Preschool X” (anonymized name)T: that sounds great! And do you think that would help?C: Mmm

The teacher invites to the talk by referring to the preschool’s theme work/project (Line 1). The talk is led forward by the teacher, who attaches follow-up questions that are linked to the child’s sayings (Lines 3, 5, 7, 9). In this way, she shows the child that she is interested in his ideas and invites him to develop them further. In lines 7 and 11, the teacher draws the attention to other people and their behavior. The child suggests making a sign (Line 8), and when the teacher accepts this idea (Line 9), she also encourages the child’s action and recognizes the child’s right to agency. The overall impression of the talk is a back-and-forth conversation between two active partners on a very important aspect of sustainability—animal protection.

In *Excerpt 2*. *Creating sense around co-use and economy*, we follow a child talk with a five-year-old, also from the beginning of the talk.

Excerpt 2

T: Can you tell and describe what the word co-use means?C: MoneyT: What more do you think?C: You don’t have to buy everything new. You can collaborate and exchange things with each other. You can borrow from other preschools, do you remember that we talked to Preschool Y (anonymized). But you know, they have received things from us, but they haven’t given us anything. Have they forgotten it or …?T: We can talk to them and ask about thatC: Yes please, can we talk to them again, on the computer. But can’t we go there some time, it’s not that far.T: What a good idea I think, I think we should do thatC: YesT: Is there anything else that can be shared?C: (thinks for a long time and then says) Books of course. You can borrow books downtown.have to buy new things all the time, I’ll tell my mom thatT: Is there anything else you want to say about co-use?C: Just that it is important to share and to collaborateT: Why should one co-use?C: Because it’s good to learn before you start school. It’s good to learn that you don’t. 

The teacher’s initial question (Line 1) is heads-on to the topic for the talk—co-use. The answer is short and broad (Line 2). Trying to narrow it down, the teacher uses a follow-up question (Line 3) that enables the child to elaborate on the answer (Line 4). This technique of using follow-up questions that shows a listening and engaged approach from the teacher is also used in lines 5 and 7. In line 9, the teacher’s open question invites the child to another round about co-using. This technique is rewarded by the child showing more knowledge (Line 10). With the “why” question in line 11, the teacher challenges the child to share more from the learning and meaning making from the theme work. All in all, the talk circles around recent learning during the sustainability project on co-using and saving money. The teacher’s questions invite the child to share his life-world, positioning the child as a competent actor showing an interest in sustainability and active participation in the local neighborhood.

### 3.2. Question and Answer, Focusing on Remembering Facts

In this approach, we note a way of communicating that follows a structure in which the teacher asks questions, and the child answers. This kind of communication is common in education when children try to figure out what the teacher is aiming at with his or her questions. Underpinning this approach lies the idea that education and learning are about passing on facts from the teacher to the child/ren. The talks about learning narrow down to checking whether the child can repeat facts and the intended knowledge. This approach is teacher centered, and the talks serve as a way for the teacher to see whether the child remember what they have done.

In the third excerpt, *Excerpt 3*. *Focusing paper and paper waste*, we meet a four-year-old girl and her teacher.

Excerpt 3

T: Now I was going to ask you what kind of paper it is in the drawer behind you?C: (sucks in air)T: What kind of paper is that?C: WoodT: Is it wood?T: How do we get our paper here to the preschool?C: Don’t know (pause) I don’t knowT: At home how do you get paper at home that you draw and write on?C: (silent) It lags a bit (a comment about the tablet the teacher is using for recording the talk)T: What happens when the paper runs out?C: Then you have wastedT: What do you do when you run out of paper then?C: (quick answer) make new paper

The talk starts when the teacher asks the child about something familiar in the preschool environment (paper) (Line 1). It takes two turns, and the child’s answers are short and broad, as in the beginning of the talk in Excerpt 2. The teacher then asks a question with a possible right or wrong answer (Line 5), and when there is no answer, the teacher changes track and asks about something else (Line 6). This was a hard question for the child (Line 7), to be followed by another, in which the teacher relates to the conditions at home (Line 8). This question is not answered; instead, the child shows that she is following the process of the tablet that is used for recording the talk (Line 9). Looking at the two next questions (lines 10 and 12), the teacher does not follow up on the child’s answers, although the child’s answers create space for interesting thoughts and ideas, about wasting paper (Line 11) and making new paper (Line 13). 

The talk follows a question-and-answer structure. Maybe the teacher had prepared the questions in advance; maybe not. Some teachers seem to have prepared the questions in advance; others have not. There is a distance between the questions and the answers that the teacher cannot bridge. There is no shared focus on the teacher-introduced topic about paper.

In *Excerpt 4 Planting and cultivating in preschool*, we meet a three-year-old child and her teacher, and the talk circles around the recent theme work of cultivation, in which one activity was for each child to plant seeds in a pot.

Excerpt 4

T: Look here NN, this is yours, right? (T brings out NN’s pot)T: If you look at the pictures here, do you remember NN what we did here? C: Cultivated.T: Yes, what have we planted?NN is silent and looks thoughtfully at T.T: Do you remember what we put in the ground?C: NoT: No. It’s perfectly okay not to remember.T: Do you remember what you have planted here? (T points to NN’s own plant)C: CucumberT: Cucumber. How nice!T: What do you think will happen to the plant when you bring it home?C: GrowT: It will grow. Will it be more than just leaves? (The plant is currently just leaves)C: CucumberT: It will be cucumber. How nice.T: NN do you think you’ve learned anything from this cultivation theme we’ve had?C: Mm maybeT: What have you learned?The child walks away

Putting seeds in a pot is a common activity in Swedish preschools. The teacher has brought some pictures from the project, and the child’s pot is nearby. The teacher starts the talk by pointing at the pot and the pictures (Lines 1 and 2). The “what” question gets a short but accurate answer (line 3). The teacher then continues with “what” questions (Lines 4 and 6), which are answered in a negative and monosyllabic way (Lines 5 and 7). The talk continues in the same way: the teacher asks “what” questions, and the child gives short answers, which are not mirrored back by the teacher. 

The type of “what” questions that the teacher uses throughout the talk lean on an expectation that there is an answer, or even a right answer. The young child shows with her answers that she does not really know what the teacher is aiming at, and she shows low involvement. She is not helped by the teacher to share her experiences from the recent theme work that linked cultivation to sustainability.

### 3.3. Following the Children

In this approach to child talks, it is common that the teachers refer to the importance of listening to children. This is also a characteristic of the first approach—the joint creation of meaning. The difference becomes noticeable when looking at how the teacher balances the initiatives made by children to the teacher-initiated focus of the child talk. In this approach—following the children—we note that the child seems to take the lead, and the teacher follows the child’s initiatives. The focus on sustainability for the child talks, or for the related theme work in the preschool, is not clearly presented. 

In the next excerpt, *Excerpt 5*. *Finding a dinosaur egg*, the child takes the initiative right from the start, and we do not know what the teacher had prepared to focus on.

Excerpt 5

C: Look what I found!T: What is it?C: A dinosaur eggT: Wow, where did you find thatC: Down there (points to the fence and the tree)T: Is there anything elseC: It was thereT: Is there anything else?C: There’s another shell…. It’s probably an ant that has lived in the eggT: do you think, yes it is quite a small egg….

This child talk starts in a different way. The child enters the room and immediately sets the agenda (Line 1). The teacher follows this initiative and just steps into the conversation (Line 2 and 4). The child points at the fence, and the teacher asks if there is anything else related to the finding (Line 6). The child introduces a shell and a theory about an ant living in the shell (Line 9). In line 10, the teacher agrees with the child. The continued talk goes on with the child leading the way. 

The teacher is focused on the child and asks questions that show involvement in the child’s story. However, there is no attempt to turn the conversation to the sustainability theme work in the preschool. Though an interesting talk, it is, however, not related to the given task to communicate with children about learning and meaning making during the recent theme work for sustainability. 

In the last excerpt, Excerpt 6. A child inviting Sleeping Beauty to the talk, the talk started focusing on food and ended up with Sleeping Beauty.

Excerpt 6

C: So we shall eat. So we live. Good that we live.T: Yes that’s good isn’t it?C: If we die not goodT: Yes, no Maybe it’s not so good. We want to live for many yearsC: And torosa is not funnyT: What’s not funny?C: It’s not fun to get torosaT: Roses?C: No, torosaT: Panties?C: No (laughs) TorosaT: What? Can you explain a bit?C: TOROSA. A princess. She sat on a thorn, then she slept for a hundred years.

The talk started around the theme work about food, and the excerpt shows what then happens. In Line 1, the talk is about food and how important food is in our lives. The child goes on to the risk of dying (Line 3), and the teacher acknowledges this (Line 4). Suddenly, the child changes focus (Line 5) and starts talking about *torosa*. In Swedish, Sleeping Beauty is called TÖRNROSA, but the child pronounces the name slightly differently as TOROSA, and the teacher does not understand what the child is saying. She asks twice for clarifications (Lines 6 and 8). She even proposes an explanation (Line 10). Panties in Swedish is *trosor*, quite close to what the child repeatedly says. The teacher asks for further explanation (Line 12), which is given by the child (Line 13). The talk then continues around Sleeping Beauty, and the original focus of the talk is lost as the teacher follows the child’s line of thoughts. 

In the talks in this approach, the teachers have a rather relaxed and attentive approach. This is shown by the teacher staying in the background and inviting initiatives from the children. The content may then turn away from the recent theme work on education for sustainable development.

## 4. Discussion

This study analyzes systematic child talks, in which teachers were given the task to communicate with one child at a time around recent theme work or projects linked to sustainability carried out in a preschool. The transformation towards a culture of sustainability in ECE, as in society at large, leans on communication and democracy [1,3,5]. Reorienting towards sustainability demands of all people the ability to express themselves as citizens, listen to others, and negotiate meaning jointly. In ECE, the promotion of communicative skills is fundamental, and correspondingly, it is important that teachers are aware of patterns of communication and the ways in which to structure systematic talks about children’s knowledge and learning [18,19]. Language is closely related to the development of the brain, and communication—verbal and non-verbal, in play and learning—is one way for teachers to approach the children’s perspectives [24,44]. 

The analyses of the 399 systematic child talks in the study, however, showed a variation in the teacher’s approaches to the task. Three different approaches were identified: (1) joint creation of meaning, (2) question and answer, focusing on remembering facts, and (3) following the children. As we have shown in the excerpts, the teachers structured the child talks differently and used different types of questions. 

In the first approach—joint creation of meaning—the talks were held in an interactive atmosphere, and the child was a collaborative partner in the dialogue. Such an approach makes proximity to the child’s life-world possible and gives rich opportunities for the teacher to build a responsive and supportive approach to children’s learning [21,28]. Education for sustainability is about empowering all learners in a transactive and transformative process, in which we need to come up with new ideas, solutions, and behaviors. In this first approach, such a creative atmosphere [27] is possible, as is the child’s right to participation in daily life [12]. 

In the second approach—question and answer, focusing on remembering facts—there is a tendency that the dialogue becomes monologic, in which the child is invited to an educative talk with the more knowledgeable teacher, who has the knowledge that is to be passed on. The question-and-answer approach is common in education, and there is a risk that children are searching for “the right answer” or trying to guess what the teacher is thinking of. This may result in an unstable atmosphere [28] for the child, and as we found in child talks in this approach, the teachers often seemed to have a view of the child as becoming. The approach gives the child less opportunities to come forward with ideas and suggestions, in this case, for building more sustainable habits. 

In the third approach—following the children—the child is looked upon as rich and competent. Anything may become interesting and play and learning meander because of this. The ability to listen to children and to strive towards the children’s perspective is considered to an indicator of quality in ECE [22,45]. However, ECE is about promoting learning and the development of democratic values, for instance, on children’s rights and sustainable development [8,12]. Teachers are to ensure teaching and learning in accordance with the tasks, goals, and guidelines set out in the national curriculum [35]. The “follow the children” approach does not guarantee that education is directed towards a common focus, a shared meaning making around sustainability. The approach opens up space for large variations in ECE quality and inequitable preschools, and it does not guarantee education for sustainability. 

These findings confirm research by Johansson [16,30]. Quality ECEC varies with the teachers’ views on children, how they look at knowledge and learning, and how these different categories become visible in their approaches to teaching and learning in ECEC. Common for two of the approaches—2. question and answer, focusing on remembering facts and 3. following the children—was weak intersubjectivity. Being able to establish intersubjectivity is a characteristic of high quality ECE [4,22]. In the second approach, an adult perspective was in the lead, and in the third approach, a child perspective was dominating the child talks. Intersubjectivity is a link to understanding the child and a way to create a shared focus during the talks.

Early childhood pedagogy aligns quite easily with the UNESCO policy around ESD [10,46], which highlights active or activity-based learning methods. The systematic child talks were centered around recent theme work or projects for sustainability. Project-based learning, in which children are actively exploring real-world problems and challenges, situational learning, in which children are involved in cooperative activities where they are challenged to use their critical thinking, and place-based learning that immerses children in local heritage, cultures, landscapes, opportunities, and experiences are examples of quality ECEfS. 

The three different approaches that were identified in this study are the result of data coming from everyday ECE practices in Sweden. They point at a large variation in teachers’ communicative competences, as exemplified by the way they performed systematic child talks. 

A limitation in this study was that the participating teachers were not randomly chosen. They all follow a research and development program on sustainability in ECE, which suggests an interest in EfS. On the other hand, they were appointed to the program by their employers, and we have no knowledge of the criteria used when choosing the participants. Most importantly, all participants signed a written consent to participate in the study. 

## 5. Conclusions

The purpose of this study was to explore the ways in which preschool teachers tackle the task of conducting systematic child talks in their practices, focusing on education for sustainable development. The analyses of the 399 child talks—the data—revealed a variation in the teachers’ approach during the talks. A key factor during the talks was the teacher’s competence to create a dialogue in a shared inter-subjective atmosphere, while at the same time being open for alterity, that is, introducing new or slightly changed perspectives for the dialogue during the child talks in order for them to deepen and to continue. 

ECEfS may easily be integrated in the daily ECE program, in which play marks the foundation [22]. Play is characterized by creativity, fantasy, communication, and negotiation, and in play, the level of child participation is high. These same characteristics are also highlighted in ECEfS. This is why ECE theme and project work are natural arenas for building cultures for sustainability. These findings could be informative and useful for both university programs for ECE teachers and professional development courses. 

To conclude, this study shows the importance of having skilled ECE teachers with the competence to communicate with children and with an interest in sharing the children’s life-world. Planning for these kinds of playing and learning activities in ECE is in the best interests of all children and will contribute to the development of a sustainable society. 

## Data Availability

The data in this study are secured in a web portal at Mälardalen University.

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
