# Peer review of "Systematic Child Talks in Early Childhood Education—A Method for Sustainability"

_children, 2023, doi:10.3390/children10040661_

Round 1

Reviewer 1 Report

Thank you to the authors for their work on this paper, which was really engaging and insightful. I was glad to have the opportunity to review.

The paper is well developed across all areas. Some minor copy editing is still required for clarity and accuracy. 

Re: "Maybe the teacher..." (lines 434-435), the tone here shifts to something which feels more casual and at odds w/ the rest of the paper. I suggest a slight revision to keep tone consistent.

Findings are particularly well presented with good depth, detail, and clarity. Great use of excerpts and comprehensive examples and explanations given across this section.

The very last sentence in the concluding section could use some work to enhance clarity and impact. For example, see the following suggested amendment: "Planning for these kinds of playful learning activities in ECE is in the best interests of all children and will contribute to the development of a sustainable society."

Overall a strong + interesting paper covering an area of great significance. Well done to the authors on their work.

Author Response

Thank you for helpful comments and for the general positive review. 

We have edited the specified lines in two sections, and thank you for the support. 

Reviewer 2 Report

Regarding the content, I do not have any changes to recommend, it makes a good literary review to support the relevance of the problem to be studied and a good structuring of the content, it uses the correct methodology for this type of study and it is a consistent and well-detailed methodology to give significance to the results they show, makes a good discussion of the results with respect to the studies carried out previously, and marks the conclusion obtained well.

Although I advise looking at these things:

Never two sections without a paragraph of text in between. You should put a couple of lines describing/naming the subsections you are going to deal with within that section. You must correct this between sections 2-2.1.

The section “4. Discussion” should be called “4. Discussions and conclusions".

In the section “4. Discussion and Conclusions”, it is necessary to develop a deeper analysis of the conclusions, implications and limitations of the study. In addition to the possible future lines of research opened with this research.

And the references in the 'References' section must follow the model set by the journal. You must correct the errors that exist. Look at this in the template.

Author Response

Thank you for your comments.

We have added some metatext between 2. and 2.1.

Your comment on the end of the article is similar to another reviewer's comments. The article now has a final section 5. Conclusion. 

Reviewer 3 Report

This article is focused on an important and exciting topic. The results obtained from the research are beneficial for creating sustainable learning in early childhood education. However, a few revisions are required. 

1. Introduction: This section is written well, and authors may consider adding more elements on the importance of teacher-student communication. 

2. Methods and Material (Literature Review):  The first few sub-headings do not describe methods and materials, but they instead provide a literature review. So, those headings are better moved to "literature review." A theoretical perspective can be placed either in Literature Review or in Methods and Material. 

3. Methods and Material (Methods context): This section requires further information about the methods and materials used to conduct the study. For example, more information about the participants (teachers and students) is required, i.e., their background information. Further, it requires detailed information on the data collection techniques. At present, there is confusion in understanding the data collection methods. Whether systematic talks were collected before training (as claimed in lines 260 - 270) or after the conference (lines 271 - 280). And, more information is required for the conference and training content for the teachers. 

It would be better if the methods and materials section was divided into sub-headings so readers could understand it easily. (for example, study context, participants, data collection techniques, data analysis, etc. )

Results and Discussion: The results are interesting, explained well,  and valuable for early childhood teachers. The discussions are focused on the results. 

Missing sections: The article has some missing areas that could be added, such as the conclusion, practical implications, limitations, and suggestions for future research. The conclusion will help readers understand the article's main findings, and practical implications will help readers to apply these findings in actual practice. Limitations and future suggestions will allow researchers to conduct more research on the topic to fill the research gap. 

Note: The title could be revised by making it more detailed so that readers could find the primary information (research topic, methods, location, participants, etc. ) about the article. 

Author Response

Thank you for valuable comments. we have addressed them :

2. WE understnad your scomments about a section on Literature review. However, we have followed the Journal's strict format, which does not include a separete section on Literture review. However, the first parts of Section 2 is our review. 

3. We have made clarifying amendments in the Methods-section.

4. The article now ends with a section 5. Conclusion, where we have written about the missing sections.